# Spatial Prioritization for Ecotourism through Applying the Landscape Resilience Model

Shekoufeh Nematollahi [1], Sadaf Afghari [1], Felix Kienast [2] and Sima Fakheran [1,*]

[1]   Department of Natural Resources, Isfahan University of Technology, Isfahan 84156-83111, Iran
[2]   ETHZ & Swiss Federal Institute for Forest, Snow and Landscape Research (WSL), Zürcherstrasse 111, CH-8903 Birmensdorf, Switzerland
*    Correspondence: fakheran@iut.ac.ir

**Abstract:** Given the growing universal demand for sustainable development in recent years, ecotourism has become one of the top effectual actions that can be employed to reconcile environmental conservation with economic growth. Therefore, sustainable development can be supported by assessing ecotourism ecosystem services at the landscape scale. In this regard, we presented a new technique that considers a potential model of ecotourism along with a landscape resilience measurement to identify the priority areas for sustainable ecotourism development. For this purpose, a multi-criteria fuzzy model with a geographic information system (GIS) and analytical hierarchy process (AHP) was first used to evaluate potential zones for ecotourism. The landscape ecological risk index (ERI) was then applied to measure the landscape resilience. The usefulness of our novel technique was then tested in a case study in the Chaharmahal and Bakhtiari province (Ch & B), situated in the central part of the Zagros Mountain Chains, Iran. The area has a coarse terrain with climate that varies considerably, which results in high potential for ecotourism development. The results indicated that about half of the provincial area had high potential for developing ecotourism and attracting tourists. However, when considering the landscape resilience, approximately 33% of the study area near the western and central regions had both high potential for ecotourism and the high values of landscape resilience, making these locations suitable for sustainable ecotourism development. Overall, the present study demonstrated that utilizing the integrated models and the ecotourism potential model, together with the landscape resilience assessment, might provide a powerful tool for ecotourism prioritization for the purpose of sustainable development.

**Keywords:** ecotourism; sustainable development; ecosystem services; landscape resilience; prioritization

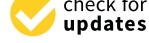



## 1. Introduction

In recent decades, there has been a growing universal need for sustainable development. As described by the International Institute for Sustainable Development (IISD) in 2012, sustainable development is growth that satisfies present requirements without undermining the capability of next generations to afford their needs [1]. One of the key issues in today's world is sustainable development through ecosystem services, particularly cultural ones, which refers to the chances provided by nature for tourism and recreation [2]. Ecotourism is a subgroup of the tourism industry that places emphasis on the maintenance and enlargement of natural systems through tourism [3,4].

The National Ecotourism Strategy (1994) described ecotourism as nature-based tourism that encompasses training and explanations regarding the natural environment and that is mainly managed to be ecologically sustainable [5,6]. Ecotourism involves the sustainable use of natural resources and also involves local people for the purpose of conserving the biodiversity and ecology of the area, while providing economic benefits for nearby communities [7,8]. Therefore, ecotourism is one of the most effective actions that can be utilized to

reconcile environmental conservation with economic development [9]. Numerous countries have ensured their local or regional development by considering sustainable development through ecotourism development [10]. Ecotourism ecosystem services should therefore be identified and then valued to ensure the sustainable development of a multi-functional landscape that can support human well-being [11]. One of the most effective planning tools for this purpose is the geographic information system (GIS), which has been used in recent research studies.

To obtain a comprehensive assessment, it is necessary to consider other parameters that affect the ecosystem's quality and the services it provides. The land use/cover changes made by humans, such as urbanization, deforestation, agricultural encroachment, and infrastructure development have the greatest negative impacts on ecosystem services [12,13]. These changes constantly create landscapes that are less resilient, expanding the anthropogenic and natural risks and affecting the quality of life [14]. Landscape resilience pertains to the capability of an environment to resist external disturbances and to reorganize itself to maintain its critical structures, functions, and mechanisms [15]. Hence, resilience is a key aspect that is gaining significance in landscape studies [16] due to its ability to define degraded landscapes and environments through urbanization and development. The analysis and evaluation of ecotourism as representative of advancing land sustainability has promoted the concept of landscape resilience [17]. Therefore, the methodological and systematic integration of the basic principles of landscape resilience and ecotourism ecosystem services can provide significant tools for landscape analysis, which contributes to the adjustment of sustainable activities in particular regions, particularly those which are inordinately anthropized [18,19].

In this regard, we present a new technique that considers a potential ecotourism model and landscape resilience measurement to identify priority areas for sustainable ecotourism development. The usefulness of our novel technique was tested in a case study in the Chaharmahal and Bakhtiari (Ch and B) province, which provides diverse ecosystem services, such as biodiversity maintenance, food provision, and water and climate regulation, as well as recreational opportunities. Based on the published reports, out of a total of 538 tourist attractions in the province, 109 were natural attractions, 325 were historical cultural attractions, 148 were religious attractions, and 1 was a health village. The number of tourists who chose the Ch and B province as a tourist destination was equivalent to 916,000 people per year, with an average cost of $20 for each tourist. Considering the special topography and mountainous conditions of the province, human access to these tourist attractions facilitated solely by road networks consumes more time and expense compared to those of flat areas. In addition to the road networks, other facilities such as distance to settlement areas as well as health and treatment centers, are considered important infrastructural facilities for ecotourism development [20]. However, irregular economic development and land use/cover changes, specifically the conversion of natural habitats to agricultural lands and built-up areas, such as urban and road networks, have caused changes in the ecosystem and in the services it provides. It is, therefore, requisite to identify the priority areas for sustainable ecotourism development. For this purpose, we integrated the potential ecotourism model and the ecological risk index, representing landscape resilience, in order to recognize appropriate locations for sustainable ecotourism development.

The questions of this research study were as follows:

- Which areas have high ecotourism potential?
- What is the condition of different areas of this province in terms of landscape resilience?
- In which areas is sustainable ecotourism development possible?

## 2. Materials and Method

### 2.1. Study Area

The present study concerned the Chaharmahal and Bakhtiari province, with an approximate area of 16,332 km² and an average annual rainfall of 560 mm, which is situated in the central section of Iran. The average temperature in the hottest and coldest months of the year are 24.6 and −5 degree Celsius, respectively. The prevailing wind direction is from the south and southwest of the province, with an average wind speed of 1.4 m/s. The air pressure is relatively low in all seasons and an average annual air humidity is 40% throughout the study region. The number of frosty days is between 83–140 days, and the sky is cloudier in the winter season compared to other seasons, with the number of cloudy days varying from 21 to 48 across different cities. About 6 to 7 days of the year are stormy with lightning in Shahrekord City, the capital city of the Ch and B province. The geological structure of this province was formed from marine sediment (Tethys) at the beginning of the third geological period, and the organic movements of the Zagros mountain range caused the appearnce of the Zagros to fold. The soil cover is very shallow in Shahrekord City, relatively deep in the mountainous and hilly areas, and very deep in the plains [21].

This case study is predicated on a coarse topography, and the climate varies appreciably, which results in a high ecosystem variety. This provides appropriate habitats for an extraordinary range of plant and animal species, including the Persian leopard (*Panthera pardus*) and wild goat (*Capra aegagrus*), which are categorized as endangered and vulnerable species, respectively, on the International Union for Conservation of Nature (IUCN) red list. This province supports around 1200 plant species and 294 animal species, including 170 bird species, 62 mammal species, 35 reptile species, 22 fish species and 5 amphibian species [21]. In addition, this province has many appealing features, such as waterfalls, forests, and wetlands, that have attracted a large number of tourists (Figures 1 and 2A). According to the land use and land cover map (LULC), produced in 2014 by the Organization of Forests, Ranges and Watershed Management, about 9.6%, 20.57%, and 55.6% of the case study area are agricultural lands, forests, and rangelands, respectively (Figure 2B). Approximately 11.5% of the Ch and B province is considered a protected area network, encompassing one wildlife refuge (Shirestan), one national park (Tang-e-Sayyad), one national, natural monument (*Fritillaria imperialis*), and five protected areas (Helen, Sabzkouh, Teng-e-Sayyad, Gheisari, and Sheyda). The central section of the case study, encompassing the Tang-e-Sayyad protected area along with the Sabzkouh protected area, was named a biosphere reserve by the Man and Biosphere Programme of UNESCO (MAB) in 2015 because of the unique endemic fauna and flora. Therefore, this province has great potential as a tourist destination, and it is necessary to recognize high-priority regions for ecotourism development. However, these natural areas have also been negatively affected by anthropogenic activities, particularly urbanization and the development of road networks, which have caused habitat loss and fragmentation. For this reason, it is crucial to recognize the priority areas where sustainable ecotourism can be established.

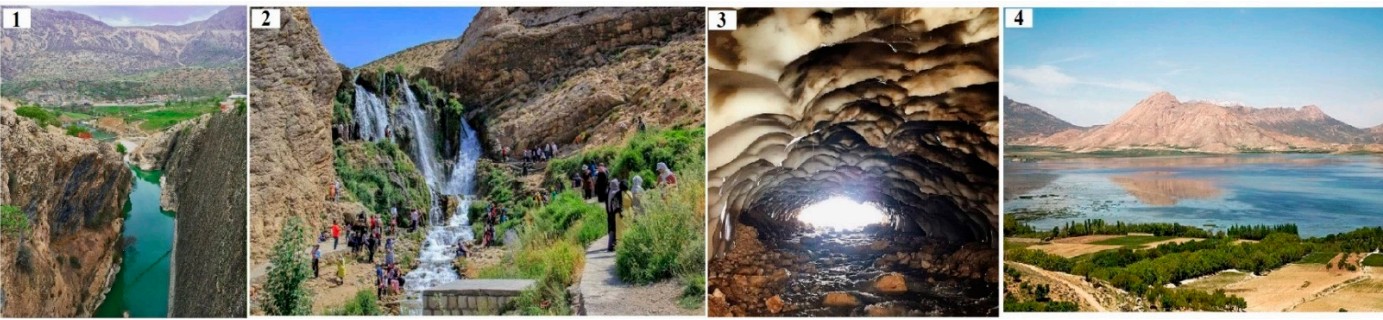

**Figure 1.** Typical Ecotourism areas across the study region; 1—Dopolan River, 2—Sheikh-Alikhan waterfall, 3—Chama Cave, 4—Choghakhor wetland.

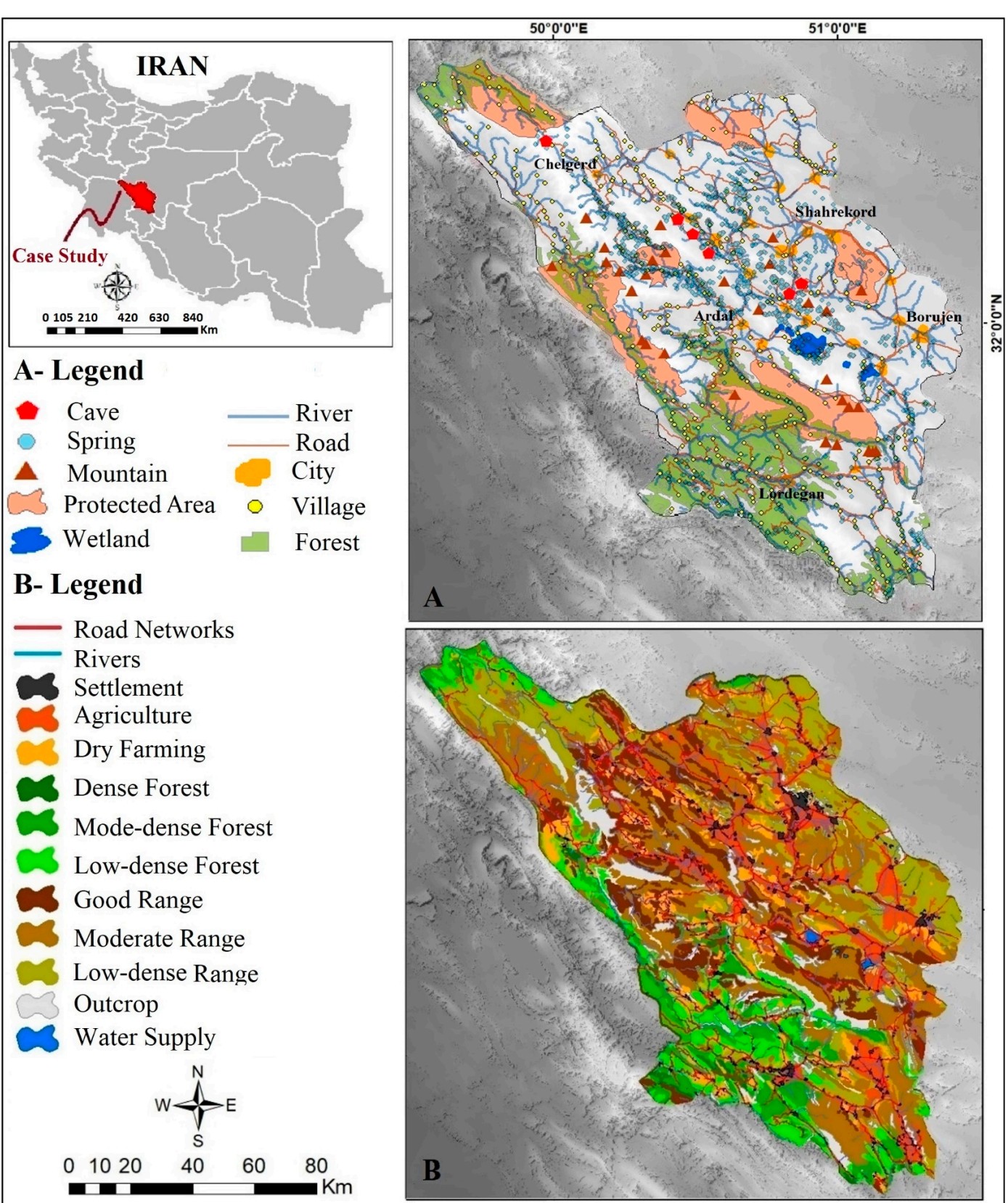

**Figure 2.** Position of the study area: (**A**) locations of the tourist attraction factors, (**B**) land use/cover map.

*2.2. Method*

In order to recognize the priority areas for the development of sustainable ecotourism, we applied three steps, as shown in Figure 3.

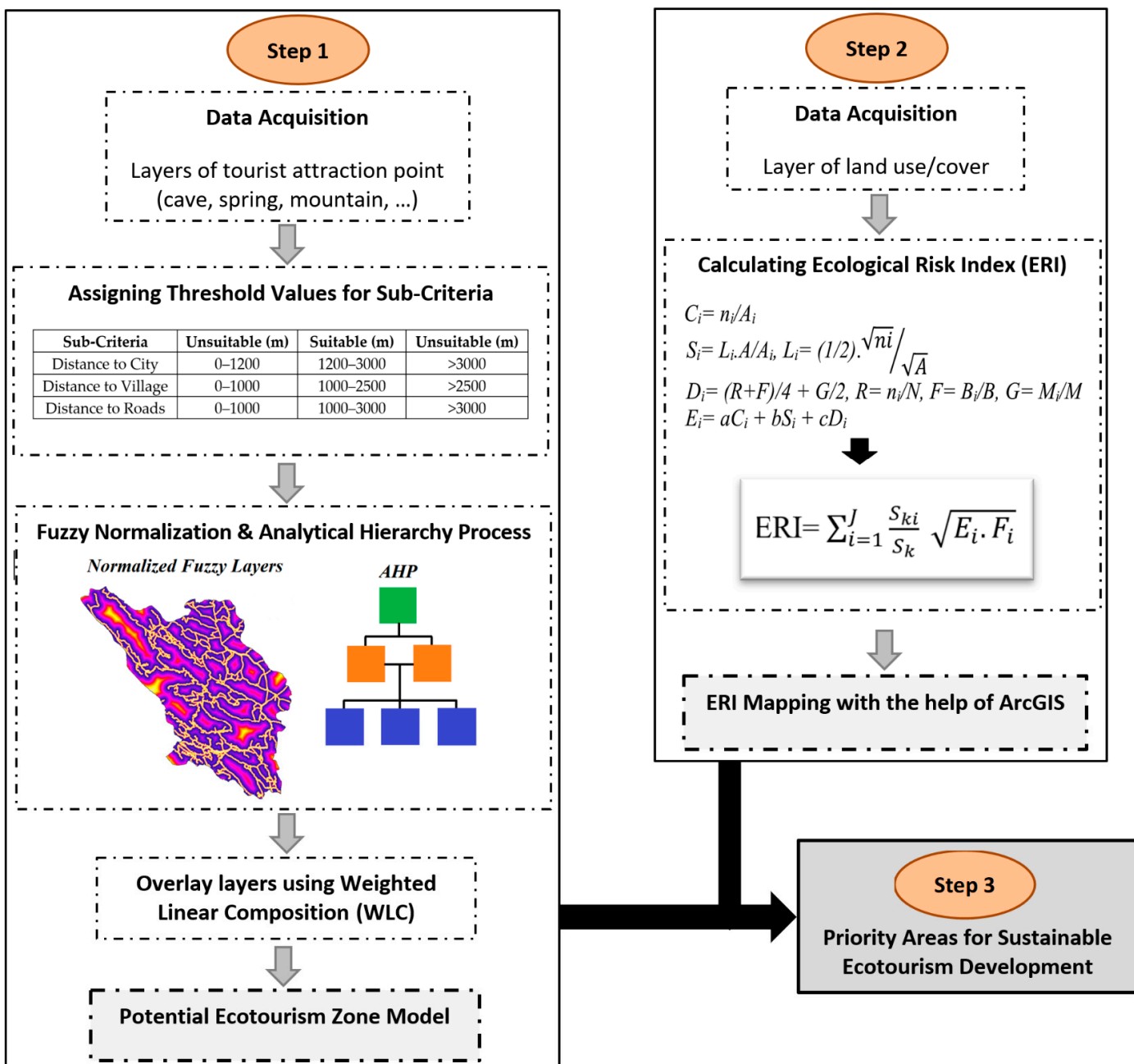

**Figure 3.** A flowchart illustrating the methodological structure of the present study.

2.2.1. Assessing and Modelling the Potential Ecotourism Zones (See Figure 3, Step 1.)

To identify the potential zones for ecotourism on the landscape scale in the Chaharmahal and Bakhtiari province, a multi-criteria fuzzy model with the geographic information system (GIS) and analytical hierarchy process (AHP) was used. First, the appropriate criteria were selected from the field visits and information based on the conditions of the study region were obtained from interviews with local experts. These were then divided into two main groups: (1) ecological criteria (protected areas, bodies of water, caves, mountains, forests, and wetlands) and (2) physical criteria (road networks, cities, and villages) (Figure 4).

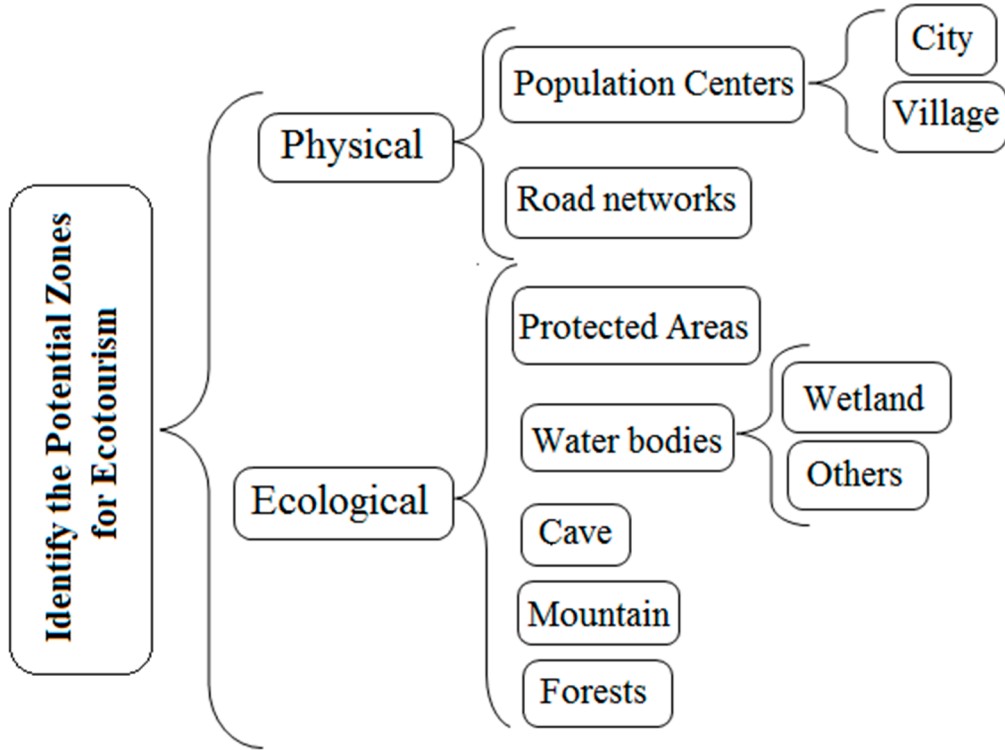

**Figure 4.** Selected criteria and sub-criteria for identifying the potential zones for ecotourism development.

After collecting the information layers of each sub-criterion, the Euclidean function was used to rasterize them with the help of ArcGIS. The fuzzy logic in the Terrset software was then used to standardize the above-mentioned layers between 0 and 1. For this purpose, it was necessary to determine the threshold quantities of the sub-criterion and the types and shape of the membership function. Therefore, the threshold values were identified according to the literature reviews and expert knowledge (Table 1).

**Table 1.** The threshold quantities of the sub-criterion and the types and shape of the membership function.

| Sub-Criteria | Unsuitable (m) | Suitable (m) | Unsuitable (m) | Function |
|---|---|---|---|---|
| Distance to City | 0–1200 | 1200–3000 | >3000 | Symmetric linear function |
| Distance to Village | 0–1000 | 1000–2500 | >2500 | Symmetric linear function |
| Distance to Roads | 0–1000 | 1000–3000 | >3000 | Symmetric linear function |
| Distance to Protected Areas | - | 0–2000 | >2000 | Decreasing linear function |
| Distance to Water Bodies | - | 0–2000 | >2000 | Decreasing linear function |
| Distance to Cave | - | 0–2000 | >2000 | Decreasing linear function |
| Distance to Mountain | - | 0–2000 | >2000 | Decreasing linear function |
| Distance to Forest | - | 0–4000 | >4000 | Decreasing linear function |
| Distance to Wetland | - | 0–2000 | >2000 | Decreasing linear function |

In the next step, the weight of the criteria and sub-criteria, which showed the importance of each criterion compared to others in order to recognize the potential zones for

ecotourism development, were determined based on expert knowledge and using the AHP and Expert Choice software [22]. Subsequently, three experts of the Department of Environment (DoE), four professors of the faculty of natural resources, and three rangers with more than 15 years of experience in the fields of ecotourism, conservation planning, and environmental assessment were asked to assign values to each criterion and sub-criterion. In advance of expert scoring, the purpose of the present study, as well as the structure and meaning of the tables to be filled in, were described in detail (the Supplementary Materials). After that, the layers were compounded utilizing the method of weighted linear composition (WLC), as follows (Equation (1)) [23]:

$$S = \sum_{n}^{i=1} W_i \cdot X_i \tag{1}$$

The WLC method involves the following stages: (1) Create a fuzzy layer of each sub-criteria, of which, each grid-cell contains an attribute value between 0 (low suitability) and 1 (high suitability) ($X_i$) (Equation (1)); (2) define the normalized weights of the target sub-criteria ($W_i$) (Equation (1)), which represents the relative importance compared to others and which was calculated using AHP and Expert Choice software; and (3) order all the cells on the output layer according to their overall score value, where higher values represent a higher degree of suitability.

2.2.2. Computation of the Ecological Risk Index (ERI) (See Figure 3, Step 2.)

The ecological risk index is representative of landscape resilience, which pertains to the capability of an ecosystem to preserve its critical structure and functions despite external interference caused by anthropogenic activities, such as the development of road networks or alterations in land use [15,24–28]. In order to calculate and spatialize the ERI, the province was subdivided into square sample segments. To determine the sample cell size, landscape configuration and composition metrics were applied in different square cell sizes (1, 1.25, 3.25, 6.25, 9.25, 12.25, and 15.25 km$^2$). After testing the spatial thresholds for each of the landscape metrics in different cell sizes, we found that the mutability of the landscape metrics did not change notably beyond 6.25 km$^2$. Therefore, the study region was subdivided into 2720 square sample segments of 2.5 km.

In the next step, the ERI, which consisted of the landscape disturbance index ($E_i$) and frangibility index ($F_i$), was calculated. The $E_i$ evaluated the size of external interference on natural habitats and was based on three landscape sub-indices of the dominance index ($D_i$), splitting index ($S_i$), and fragmentation index ($C_i$) (Equations (2)–(4)). In order to compute $E_i$, according to prevous studies, the weights of 0.2, 0.3, and 0.5 were allotted to $D_i$, $S_i$, and $C_i$, respectively, and then summed together (Equation (5)) [24–26,29,30].

$$C_i = n_i / A_i \tag{2}$$

$$S_i = L_i \cdot A / A_i, \ L_i = (1/2) \cdot \sqrt{ni} / \sqrt{A} \tag{3}$$

$$D_i = (R + F) / 4 + G/2, \ R = n_i/N, \ F = B_i/B, \ G = M_i/M \tag{4}$$

where $N$ is the whole number of patches, $n_i$ is the number of landscape type $i$'s patches, $A$ is the entire area of the study region, $A_i$ is the entire area of the landscape type $i$, $L_i$ is the distance index of the landscape type $i$, $B_i$ is the sample number of patches $i$, $B$ is the whole number of samples, $M_i$ is the area of the patch type $i$ and $M$ is the entire area of all samples.

$$E_i = aC_i + bS_i + cD_i \tag{5}$$

Since external interference changed the ecosystem's structure and function, the pattern of the landscape altered. Hence, the index of frangibility degree ($F_i$), which quantified the inner capability of the landscape type to retain its balance in response to external tensions, was assigned in accordance with the local conditions and expert knowledge [31]. For this purpose, the region was first segmented into six key categories of landscape type. After

that, contemplating the condition of the study region, and based on specialized knowledge, the $F_i$ was allotted to each kind of landscape from low values (1 = most resilient) to high values (6 = least resilient), i.e., to bodies of water, forests, grasslands, outcrops, farmlands, and construction lands [30]. Then, the landscape frangibility index was obtained after normalizing these indices. Finally, the ERI was calculated using the Equation (6) as follows:

$$\text{ERI} = \sum_{i=1}^{J} \frac{S_{ki}}{S_k} \sqrt{E_i \cdot F_i} \tag{6}$$

where $N$ is the number of landscape types in each sample areas, $S_k$ is the whole area of the sample area $k$, and $S_{ki}$ is the area of the landscape type $i$ in a sample area.

### 2.2.3. Identifying Priority Areas for Sustainable Ecotourism Development

To achieve the purpose of sustainable ecotourism development, it was essential to identify priority areas that had a high potential for ecotourism, on the one hand, and a high value of landscape resilience on the other. Hence, to identify these locations for sustainable ecotourism development, the layer of the ecological risk index, representing landscape resilience, was overlaid with that of potential ecotourism zones with the help of ArcGIS.

## 3. Results

### 3.1. Assessing and Modelling the Potential Ecotourism Zones

The results obtained through assessing the relative importance of the criteria, revealed that the ecological criteria with a value of 0.845 were more important compared to the physical criteria (Table 2). In addition, the protected areas, water resources, wetlands, and forest lands were more important in the ecotourism analysis, with values of 0.27, 0.174, 0.133, and 0.118, respectively (Table 2). The resulting maps, obtained using the fuzzy logic in Terrset software in order to standardize the input layers between 0 and 1, are shown in Figure 5.

**Table 2.** The weights of the criteria and sub-criteria.

| Criteria | Weight | Sub-Criteria | Weight |
|---|---|---|---|
| Physical | 0.155 | Road networks | 0.092 |
| | | City | 0.017 |
| | | Village | 0.045 |
| Ecological | 0.845 | Protected areas | 0.27 |
| | | Water bodies | 0.174 |
| | | Wetland | 0.133 |
| | | Cave | 0.079 |
| | | Mountain | 0.072 |
| | | Forest | 0.118 |

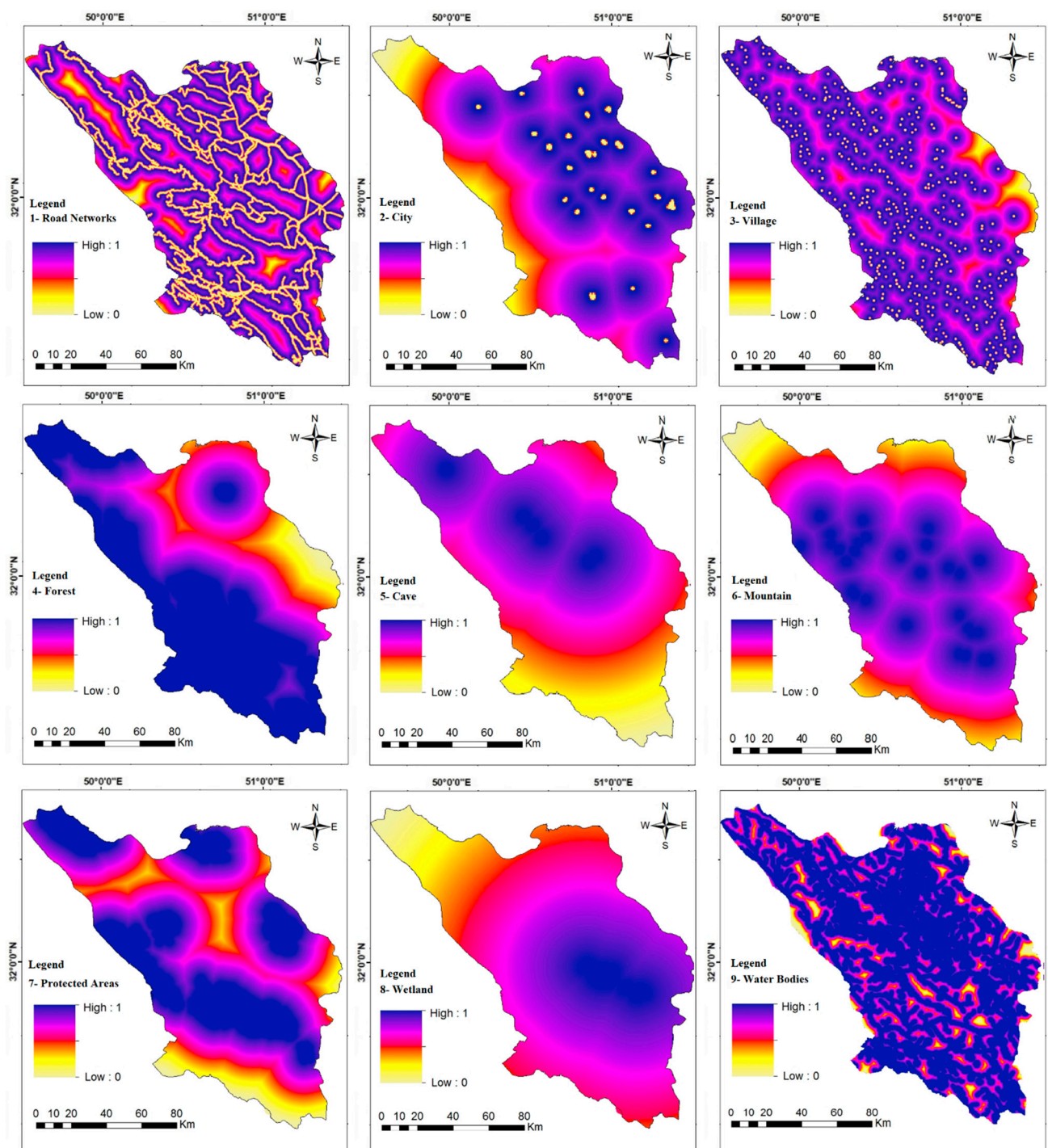

**Figure 5.** Fuzzy maps of 1—road networks, 2—cities, 3—villages, 4—forests, 5—caves, 6—mountains, 7—protected areas, 8—wetlands, and 9—bodies of water; high values in dark purple represent the high quality for ecotourism development, low values in cream represent the low quality along the different sub-criteria.

After combining the standardized data layers in ArcGIS using the WLC method, the final map was categorized into five classes utilizing the method of natural break through ArcGIS (Figure 6): extremely unsuitable, unsuitable, medium, suitable, and extremely suitable. The results showed that the eastern section of the study region around the Tang-e-Sayyad protected area and the central parts around the Sabzkouh and Helen protected areas had a high potential for ecotourism development. In addition, the Gheisari protected

area and Choghakhor wetland also offered a high potential as ecotourism development sites (Figure 6). Lower values were found for the northwestern parts around the city of Chelgerd, the eastern section around the city of Borujen, and the southern region of this province.

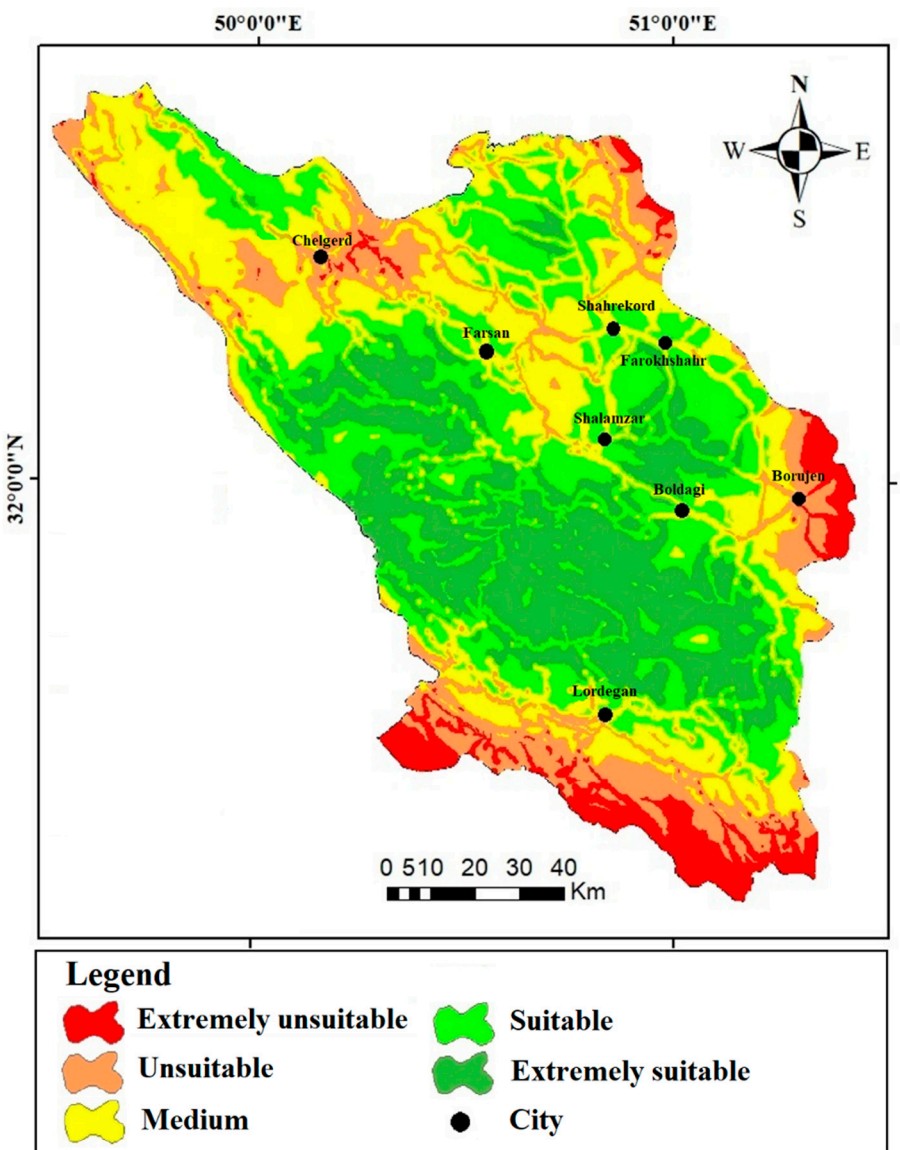

**Figure 6.** Classification map of the potential zones for ecotourism development.

According to the results, 23% of the province area (equivalent to 375,000 ha), approximately 28% (equivalent to 464,000 ha), and approximately 6% (equivalent to 105,000 ha), were placed in the extremely suitable, suitable, and unsuitable categories for ecotourism development, respectively. Indeed, the outcomes of this part of the present study demonstrated that about half of the area of the Chaharmahal and Bakhtiari province had a high potential for developing ecotourism and attracting tourists.

### 3.2. Landscape Index of Ecological Risk (ERI)

In order to analyse the landscape index of ecological risk, after implementing an ordinary, kriging interpolation with the data layer of the central point, the final ERI layer was categorized into five classes using the method of natural break through ArcGIS (Figure 7). The result indicated that ERI values were distributed unevenly throughout the study region. Higher values of the ERI were found in locations dominated by urban settlements and

dense road networks, as was witnessed in the towns of Farokhshahr, Shahrekord, Boru-jen, and Lordegan in the central, eastern, and north-eastern sections of the study region. Around 2%, 7%, and 18% of this province area, equivalent to 173 km$^2$, 1054 km$^2$, and 3030 km$^2$, were considered high, sub-high, and medium, respectively. In contrast, lower ERI values were found in the western, southern, and north-western sections of the study region, which were prevailed by mountainous areas, surrounding grasslands and forests, and a low degree of urbanization. Approximately 25,519 km$^2$ and 2654 km$^2$ of the province were in the low- and relatively low-risk categories, respectively.

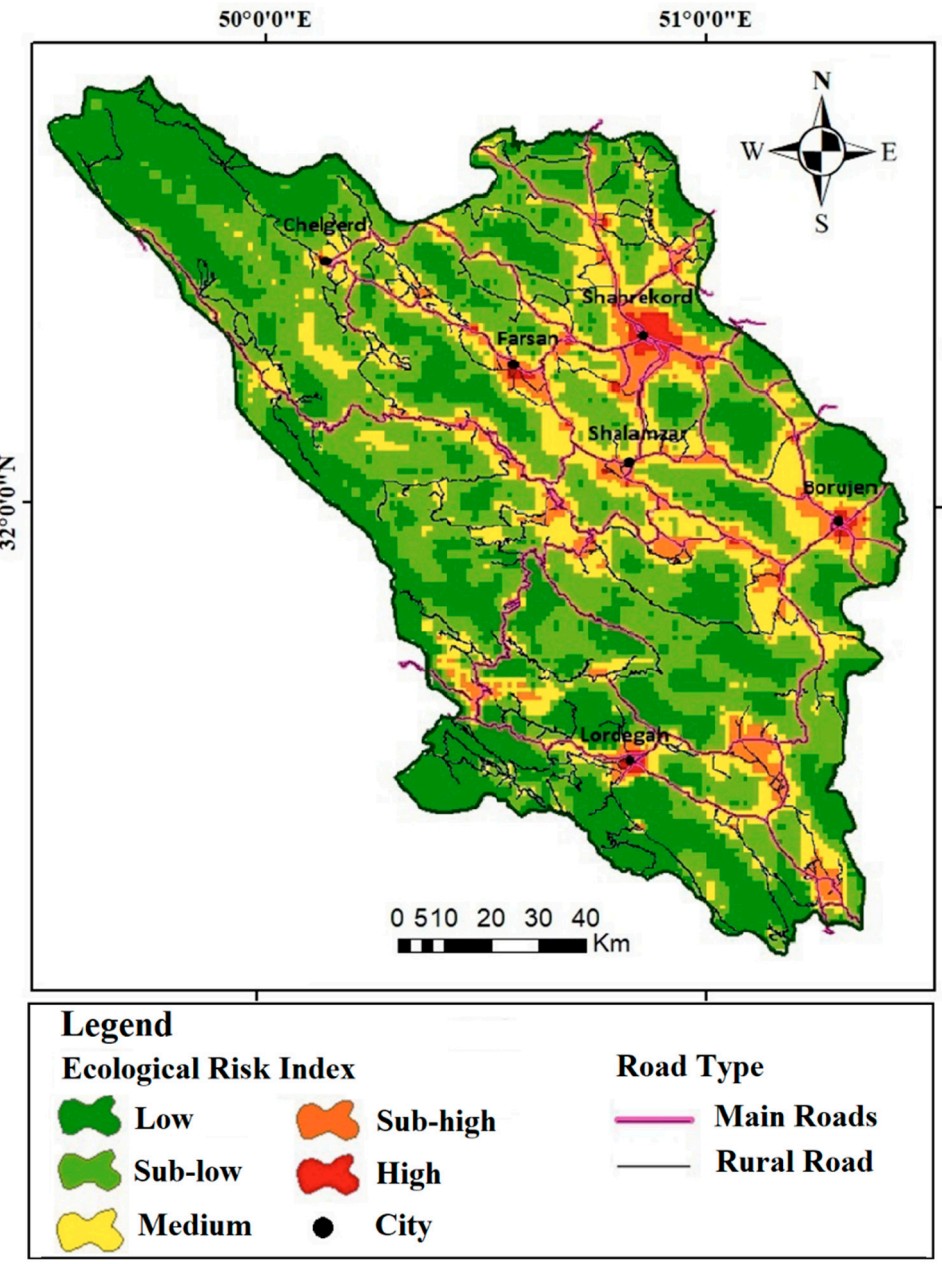

**Figure 7.** The final classification map of the landscape index of ecological risk (ERI).

In order to investigate the ecological risk status in the protected areas of the province, the final layer of ecological risk was overlaid with the protected area network layer and that of the Tang-e-Sayyad-Sabzkouh biosphere reserve (Figure 8). The results showed that approximately 15% of the Tang-e-Sayyad protected area, equivalent to 243 km$^2$, was in the high-risk category. Approximately 8%, 5%, and 8% of the protected areas of Helen, Sabzkouh, and Sheyda, respectively, were classified as high-risk. Overlaying the ecological

risk index layer with the Tang-e-Sayyad-Sabzkouh biosphere reserve layer showed that 98% of the area of the Tang-e-Sayyad and Sabzkouh core zones were in the low- and sub-low-risk categories (Figure 8). Approximately 34%, 46%, 15%, and 5% of the buffer zone area, equivalent to 853 km², 1129 km², 347 km², and 98 km², were found to be in the low-, sub-low-, medium-, and sub-high-risk categories, respectively (Figure 8).

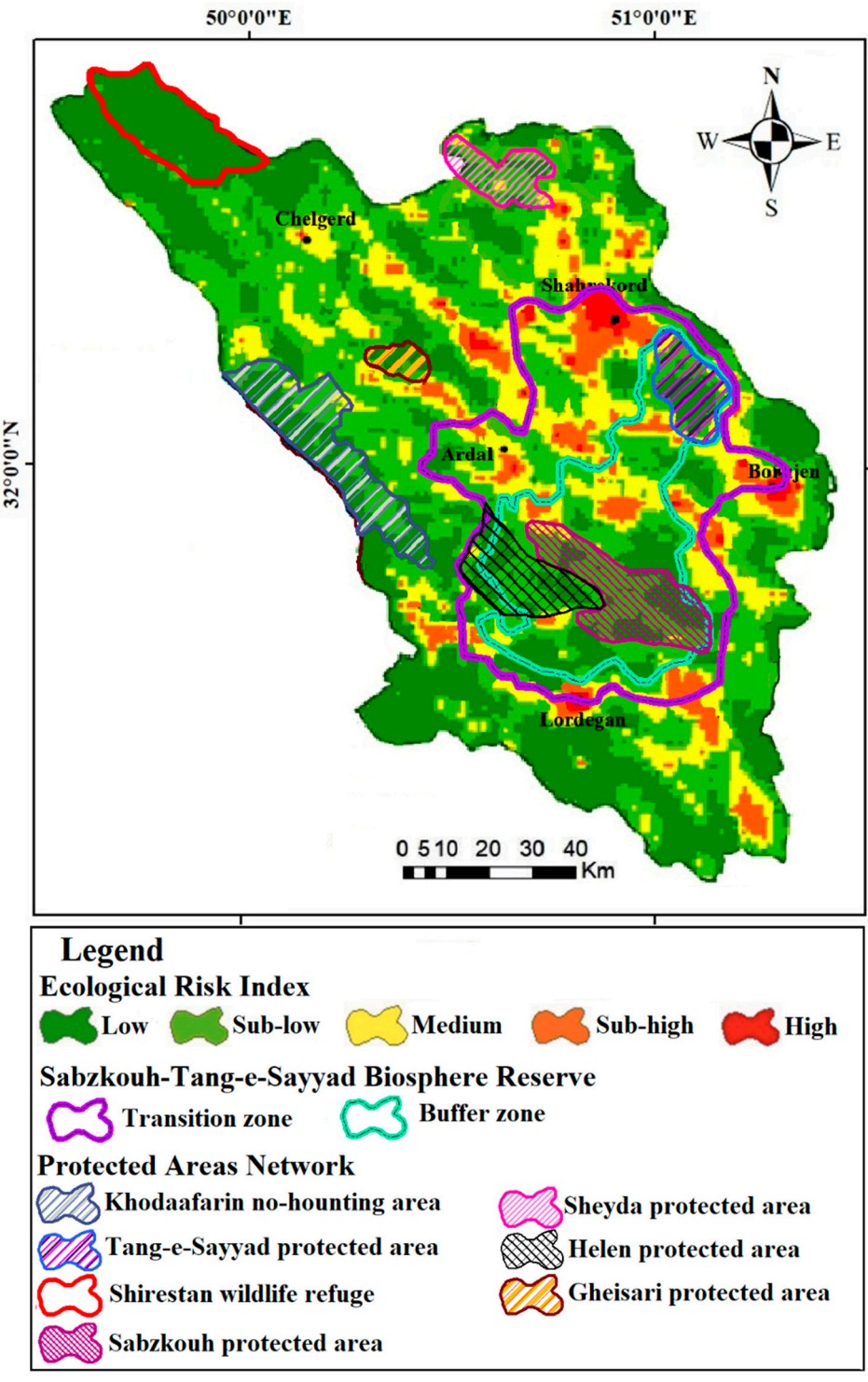

**Figure 8.** The map overlay of the ERI and protected area network layers.

*3.3. Identifying Priority Areas for Sustainable Ecotourism Development*

In order to identify the priority areas for sustainable ecotourism development across the study region, the map resulting from overlaying the ERI and potential ecotourism zones layers was divided into three zones: low suitability for ecotourism, high suitability for ecotourism–high ecological risk, and high suitability for ecotourism–low ecological risk (Figure 9).

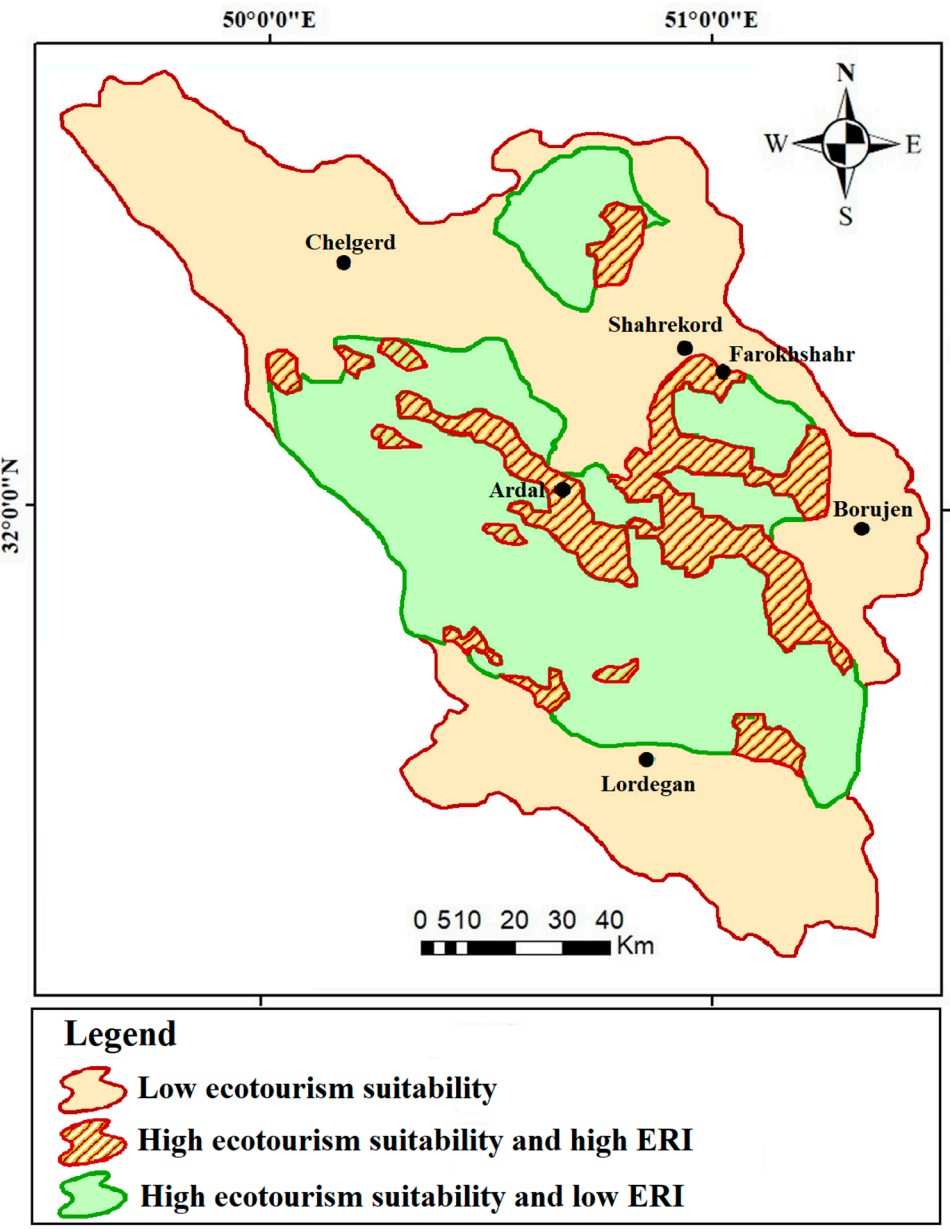

**Figure 9.** Priority areas for sustainable ecotourism development.

According to the result, although the northwestern regions had the lowest ecological risk and highest landscape resilience, these locations had the smallest potential for ecotourism due to the long distance to road networks and settlements. The central parts around the city of Ardal and the eastern parts near the Tang-e-Sayyad protected area and city of Farokhshahr, which had a high potential for ecotourism, showed lower values of landscape resilience. This was mainly owing to the fast transportation and urban developments, which had caused a more severe degree of landscape disconnection and fragmentation.

The western regions around the Khodaafarin no-hunting area, the central section around the protected areas of Helen and Sabzkouh, and the northeastern parts around

the Sheyda protected area had both a high potential for ecotourism and a high value of landscape resilience, which made these locations extremely suitable for sustainable ecotourism development. Overall, the results of this section indicate that, although about half of the province area had high potential for developing ecotourism, considering the landscape resilience and ecological risk, approximately 33% of the study area are suitable for sustainable ecotourism development.

## 4. Discussion

The present study was among the first efforts to recognize priority areas for sustainable ecotourism development in the Chaharmahal and Bakhtiari province. For this purpose, the potential zones for ecotourism were first identified using the multi-criteria fuzzy model with the help of ArcGIS. The results showed that the ecological criteria were more important compared to the physical criteria for identifying potential zones for ecotourism. The study conducted by Mafi et al. (2012) assessed the ecotourism potential of the Ch and B province based on identifying the ecotourism attractions and social-economic factors, mainly, the infrastructure criteria [32]. The methods of the studies conducted by Ghanadkar (1999) and Naderi et al. (2009) were similar to that of the present paper. However, the results of these investigations showed that a water resource was an essential criterion for identifying the priority areas for ecotourism development [33,34]. In the research performed by Bunruamkaew and Murayama (2011), GIS and the AHP were used to recognize the priority areas for sustainable tourism in Thailand [35]. Gigovic et al. (2016) employed a multi-criteria model to recognize the appropriate locations for ecotourism development in Serbia for the purpose of decreasing the negative impacts of mass tourism [36].

The final map of the potential ecotourism zones revealed that the eastern and central parts of the study region had a high potential for ecotourism development. This was primarily due to the existence of the protected areas and wetlands, which are important for attracting tourists and developing ecotourism. Another reason was the existence of the large number of caves, mountains, waterfalls, and springs near the central and northern sections of the study region. The research conducted by the Department of the Management and Planning of the Ch and B (2018) indicated that the central and northern parts of this province had high ecological potential for ecotourism development. In addition, most of the services and welfare in the province were found in these locations, which was consistent with the results of the present investigation [20].

In contrast, areas with lower values for ecotourism potential were found in the north-western sections of the Ch and B province. Although tourist-attraction factors such as forests, water bodies, and protected areas were located in these regions, the existence of limiting aspects, including the long distance to cities and population centres and the small number of roads, meant that ultimately, the ecotourism potential decreased in these areas. Lower values were also found in the southern parts due to the long distance from the protected areas, as well as the high density of the road network and villages. The existence of population centres and road networks, on the one hand, facilitated access to the areas, and had positive effects, increasing the possibility for the ecotourism development. On the other hand, population centres and road networks were considered limiting factors. For this reason, locations up to a certain distance from the population centres and road networks fell in the unsuitable and medium categories. Some parts of the Ch and B province, including the marginal areas, were considered to have a very low ecotourism potential due to the high density of roads and population centres and the lack of tourist attractions.

In general, the results of this part of the present study demonstrated that about half of the area of the Ch and B province had a high potential for developing ecotourism and attracting tourists due to its unique topography and climate, abundant water resources, and pristine and untouched landscape, which doubled the importance of designing and implementing ecotourism development plans in the province. Of course, achieving this important goal requires careful planning and the consideration of solutions, such as the development and improvement of infrastructure and facilities, the construction of perma-

nent and temporary accommodation centres in vulnerable areas, the expansion of proper communication and information, and national conferences and meetings with the experts and local people in order to recognize and expand the industry of ecotourism in the Ch and B province.

To explore whether landscape resilience was among the most critical factors for the purpose of the sustainable ecotourism development, in the second part of the present study, the ecological risk index (ERI), representing landscape resilience, was assessed with the help of ArcGIS. The results indicated that the higher values of ERI were found in areas with prevalent urban settlements and dense roads in the central, eastern, and north-eastern sections of the study region. This was essentially owing to the fast transportation and urban development, which caused a more severe degree of landscape disconnection. In fact, the road networks as well as the urban and industrial areas changed the pattern of the landscape and consequently increased its fragmentation. Due to land use changes, the resistance of land features in these areas to maintaining their balance and stability in the face of external disturbances and interventions was greatly reduced, which increased its vulnerability and ultimately expanded the ecological risk to the landscape.

In contrast, lower ERI values were found in the western, southern, and north-western sections of the study region, which was prevailed by mountainous areas, surrounding grasslands and forests, and a low degree of urbanization. Indeed, the landscape pattern, including fragmentation in the rangelands and forests of the province, exhibited a high stability. In addition, since the rangelands and forests are natural ecosystems with a large fauna and flora diversity, they had a high internal stability, such that, in the face of any disturbance and intervention, they quickly returned to their original and stable state.

According to the results, there was a close relationship between the severity of the ecological risk to the landscape and the distance from locations with a high density of road network and urban areas. The findings were compatible with those of studies conducted by Gong et al. (2015), who spatially assessed the ecological risk of the landscape in China, Mo et al. (2017), who evaluated the effects of road network development on the ecological risk of the landscape in Beijing, and Mann et al. (2020), who noted the spatio-temporal alterations in the ecological risk of the landscape in the central sections of the Himalayas [24–26].

In general, the results showed that road networks, urban- and industrial-areas were the critical factors for fragmentation in the landscape. Many studies, including those of Makki et al. (2013), who evaluated the ecological effects of the western Isfahan bypass on Iranian deer and ram species, Patru-Stupariu et al. (2015), who examined the degree of fragmentation of land features in the southern regions of Romania, and Marull et al. (2018), who assessed the ecological impacts of land use change and the development of road networks in the United States, have emphasized the negative effects of both urban and industrial areas and road networks [37–39].

In the last step, in order to identify the priority areas for the sustainable ecotourism development, the layers of the potential ecotourism zones and ERI were overlaid. The result indicated that, although about half of the province area had high potential for developing ecotourism, considering the landscape resilience and ecological risk, around 33% of the study area in the western, central, and northeastern parts are highly suitable for sustainable ecotourism development. The study conducted by Nematollahi et al. (2022), which applied Marxan—a systematic conservation planning tool—for spatial prioritization and the optimization of protected areas in the Ch and B province, showed that the western parts of this province around the Khodaafarin no-hunting area had a high priority for conservation [30]. Therefore, considering the results of the present analysis, the aforementioned locations had a high potential for sustainable ecotourism development, and these regions could be assigned as a second biosphere reserve in this province. These locations are suitable for practices compatible with environmental activities, which could strengthen scientific research and education and increase the degree of social and economic values in the Ch and B province. Indeed, these locations could improve the sustainable use of

natural resources and involve local people to conserve the biodiversity and ecology of the area, while bringing economic benefits for nearby communities.

## 5. Conclusions

The current study presented a novel technique which utilized integrated models and methods to identify the priority areas for sustainable ecotourism development. These regions had both the greatest potential for ecotourism and the highest values of landscape resilience. For this purpose, the multi-criteria fuzzy model was used with the help of ArcGIS to map the potential zones for ecotourism. The ecological risk index was then applied to ascertain the overall landscape resilience throughout the Ch and B province. These two kinds of input layers complemented one another by supplying worthwhile data in order to recognize appropriate locations for sustainable ecotourism development. There has been a lack of studies on the importance of the landscape resilience assessment in the sustainable ecotourism development. Therefore, this study exemplified the advantages of considering the ecotourism potential model, along with the landscape resilience measurement, in order to prioritize sustainable ecotourism.

One of the expectations is to consider other kinds of tourist attractions, such as historical cultural and religious attractions, as well as landownership and land cost data, in addition to other anthropogenic stresses, as a step towards rectifying the determination of potential zones for sustainable ecotourism development. Furthermore, more detailed research is required on a finer scale in the priority areas to evaluate other ecosystem services and estimate yearly economic benefits for the local societies.

**Supplementary Materials:** The following supporting information can be downloaded at: https://www.mdpi.com/article/10.3390/land11101682/s1, Figure S1: title: Selected criteria and sub-criteria for identifying the potential zones for ecotourism development, Table S1: title: Scale of the relative importance; Table S2: title: Matrix of pairwise comparisons of criteria; Table S3: title: Pairwise comparison of physical and ecological criteria; Table S4: Pairwise comparison of sub-criteria of physical criteria; Table S5: Pairwise comparison of population centers' sub-criteria; Table S6: title: Pairwise comparison of the ecological sub-criteria.

**Author Contributions:** Conceptualization, S.N. and F.K.; methodology, S.N. and S.A.; software, S.N. and S.A.; validation, S.N., S.A. and S.F.; formal analysis, S.N., S.A. and S.F.; investigation, S.N. and S.A.; resources, S.N. and S.A.; data curation, S.N.; writing—original draft preparation, S.N. and F.K.; writing—review and editing, S.N., F.K. and S.F.; visualization, S.N. and F.K.; supervision, S.N., F.K. and S.F.; project administration, F.K. and S.F.; All authors have read and agreed to the published version of the manuscript.

**Funding:** This research received no external funding.

**Acknowledgments:** We would like to express our appreciation to Ali Jafari, Saeid Pourmanafi, and the Department of Environment (DoE), specifically Ghadir Valipour, Tooraj Reisei, and the other rangers for their worthwhile inputs and help in this case study.

**Conflicts of Interest:** The authors declare that they have no known competing financial interests or personal relationships that could influence the work reported in this paper.

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
