# Peer review of "Spatial Prioritization for Ecotourism through Applying the Landscape Resilience Model"

_land, doi:10.3390/land11101682_

Round 1

Reviewer 1 Report

The paper addresses an important issue of sustainable land use governance – the prioritization of areas for ecotourism development, which is achieved by integrating the model of ecotourism potential based on various physical and ecological criteria with the assessment of landscape resilience. The applied method is scientifically sound and described in detail, although a number of points need to be clarified.

General concept comments:

1) In the introduction, as well as in the method and result section headings, the authors state that they have applied ecotourism ecosystem services value model for assessing ecotourism potential. However, the criteria proposed for this assessment (i.e., distance to physical features - city, village and roads, and to ecological features – protected areas, water bodies, caves, mountains and forests) in my opinion cannot be regarded as assessment of ecosystem service value, but rather as estimation of one aspect of recreational ecosystem services - accessibility. Therefore, I would suggest reconsidering the use of the terminology in this context.     

2) The method for identification of priority areas for sustainable ecotourism development (section 2-2-3) is not sufficiently elaborated. Consequently, also the resulting map overlaying the ecological risk index (ERI) and potential ecotourism zones layers (Figure 8) lucks justification as to how the line marking the Potential Ecotourism Zone (which probably should have been called – the Priority Ecotourism Zone) was drawn. In order to make the results more transparent and robust, I would suggest that the resulting map should perhaps divide the study area into the following zones: low suitability for ecotourism; high suitability and high ERI; high suitability and low ERI.

3) In the discussion, authors partly repeat or slightly the results of the ecotourism priority areas mapping. To avoid replication, I would recommend consolidating this information in the results section, while in the discussion to focus more on applicability of the results and the advances and limitations of the presented method.  

Specific comments:

Lines 11-12: I suppose the sentence is a bit exaggerated. Consider rephrasing it, perhaps to say that  “sustainable development can be supported by assessing ecotourism ecosystem services at landscape scale”

Line 38: Please provide reference for the National Ecotourism Strategy (1994)

Lines 44-46: It is doubtful that ecotourism could be incompatible with regional development.  Consider reformulating the sentence.

Line 51: This is a questionable statement, since land use changes depending on their character can have either negative or positive impact to landscape resilience. Maybe context should be specified.

Line 82: Abbreviations (e.g., “Ch” and “B”) first should be indicated together with the full name.

Line 93: There is mistake in the caption of the figure – the map A shows land use/cover and B – locations of the tourism attraction factors.

Line 97: The text in the Figure 2 partly is not readable. And please avoid using acronyms (e.g. WLC), which have not been explained before in the text.  

Table 1: Indicate the unit of measurement used to determine the distance; Replace “Rural” with “Village”

Lines 132-133: “in accordance with the local condition of the landscape pattern” – this could be better explained or reformulated.

Lines 139-140: Please, explain how the weights of 0.2, 0.3 and 0.5 were obtained?

Lines 176-177: This sentence is confusing – since before it was stated that

Line 190: I would suggest enriching the caption figure 4 by providing an explanation of what the fuzzy maps represent. Figures should be self-explanatory

Lines 297-198: Please, explain what is meant with “the highest level of the ecological process”

Lines 325-326: The sentence could be more elaborated.

Lines 334-335: I don’t see how this sentence from conclusions is based on the assessments or interpretations of the results provided in the paper.

Author Response

                                                                                     Isfahan, 14 September 2022

Dear Reviewer,

We are very happy to resubmit the revised version of the manuscript entitled “Spatial Prioritization for Ecotourism Through Applying the Landscape Resilience Model”.

We appreciate you for your valuable comments and suggestions which improved the manuscript considerably. We have taken all of the comments seriously and added new text to the manuscript as requested. Minor changes such as the correction of typos are not listed. We also sent the paper to a professional language review in England.

The revised manuscript has been prepared according to all the comments and views of the reviewer in green color highlight and is attached as a “Revised Manuscript”. We have also included a point-by-point response to your comments in addition to making the changes described above in the manuscript.

We hope this current version would be suitable for publication in the Journal of Land, and please do not hesitate to contact us if any further information would be required.

Reviewer 1:

The paper addresses an important issue of sustainable land use governance – the prioritization of areas for ecotourism development, which is achieved by integrating the model of ecotourism potential based on various physical and ecological criteria with the assessment of landscape resilience. The applied method is scientifically sound and described in detail, although a number of points need to be clarified.

General concept comments:

1) In the introduction, as well as in the method and result section headings, the authors state that they have applied ecotourism ecosystem services value model for assessing ecotourism potential. However, the criteria proposed for this assessment (i.e., distance to physical features - city, village and roads, and to ecological features – protected areas, water bodies, caves, mountains and forests) in my opinion cannot be regarded as assessment of ecosystem service value, but rather as estimation of one aspect of recreational ecosystem services - accessibility. Therefore, I would suggest reconsidering the use of the terminology in this context.

# Done as suggested

2) The method for identification of priority areas for sustainable ecotourism development (section 2-2-3) is not sufficiently elaborated. Consequently, also the resulting map overlaying the ecological risk index (ERI) and potential ecotourism zones layers (Figure 8) lucks justification as to how the line marking the Potential Ecotourism Zone (which probably should have been called – the Priority Ecotourism Zone) was drawn. In order to make the results more transparent and robust, I would suggest that the resulting map should perhaps divide the study area into the following zones: low suitability for ecotourism; high suitability and high ERI; high suitability and low ERI.

# Done as suggested: Line 262-264, 272-274. Fig. 9.

3) In the discussion, authors partly repeat or slightly the results of the ecotourism priority areas mapping. To avoid replication, I would recommend consolidating this information in the results section, while in the discussion to focus more on applicability of the results and the advances and limitations of the presented method.

# Done as suggested: Line 342-344, 350-352

Specific comments:

Lines 11-12: I suppose the sentence is a bit exaggerated. Consider rephrasing it, perhaps to say that “sustainable development can be supported by assessing ecotourism ecosystem services at landscape scale”

# Done as suggested

Line 38: Please provide reference for the National Ecotourism Strategy (1994)

# Done as Suggested

Lines 44-46: It is doubtful that ecotourism could be incompatible with regional development. Consider reformulating the sentence.

# Done as suggested: Line 45-46

Line 51: This is a questionable statement, since land use changes depending on their character can have either negative or positive impact to landscape resilience. Maybe context should be specified.

# Done as suggested: Line 51-52

Line 82: Abbreviations (e.g., “Ch” and “B”) first should be indicated together with the full name.

# Thank you very much for your attention. It was mentioned in line 63 in the Introduction section, for the first time.

Line 93: There is mistake in the caption of the figure – the map A shows land use/cover and B – locations of the tourism attraction factor.

# Done as suggested.

Line 97: The text in the Figure 2 partly is not readable. And please avoid using acronyms (e.g. WLC), which have not been explained before in the text.

# Done as suggested

Table 1: Indicate the unit of measurement used to determine the distance; Replace “Rural” with “Village”

# Done as suggested

Lines 132-133: “in accordance with the local condition of the landscape pattern” – this could be better explained or reformulated.

# Done as suggested: Line: 163-167

Lines 139-140: Please, explain how the weights of 0.2, 0.3 and 0.5 were obtained?

# Done as suggested: Line 170-171

Lines 176-177: This sentence is confusing – since before it was stated that

# Done as suggested

Line 190: I would suggest enriching the caption figure 4 by providing an explanation of what the fuzzy maps represent. Figures should be self-explanatory

# Done as suggested

Lines 297-198: Please, explain what is meant with “the highest level of the ecological process”

# Done as suggested: The sentence was reformulated which is more transparent and easier to understand. Line 328

Lines 325-326: The sentence could be more elaborated.

# Done as suggested: Line 350-352

Lines 334-335: I don’t see how this sentence from conclusions is based on the assessments or interpretations of the results provided in the paper.

# Done as suggested: Deleted

Reviewer 2 Report

1. It is a very interesting topic to identify priority areas for sustainable ecotourism development through ecotourism potential model and landscape resilience measurement.

2. In the "Abstract" section,it is suggested to moderately expand the research conclusions..

3. In the "Introduction" section, It is suggested to supplement the utility of ecotourism to tourists and their willingness to pay, as well as related research literature on tourists' resources and facilities in ecotourism areas, so as to provide more basis for ecotourism potential model indicators.

4. In the part of "2-1- Study Area", it is suggested to supplement climatic data such as air pressure, temperature, humidity, wind direction and speed, precipitation, thunderstorms, fog, radiation, cloud cover and cloud shape, as well as soil, geology, plants, animals and main crops, so as to provide sufficient background information for eco-tourism research. In addition, "Fig. 1." correctly places the scale position and improves the resolution of the drawing.

5. If possible, provide real photos of typical Eco-tourism areas in the study area so that readers can better understand them.

6. In the section of "3- Results", it is suggested to adjust the map layout moderately, so that Figures 5, 6, 7 and 8 are of the same size, which is easy to compare and read.

7. It is suggested to add a "discussion"section. Based on the existing research results and existing research literature, analyze, discuss and extrapolate 2-3 topics to put forward the basis for the research conclusion.

8. Research conclusions should be generalized and of general significance on the basis of case studies. Suggestions to deepen.

9. It is also suggested to supplement the research deficiencies and further research directions.

Author Response

                                                                                     Isfahan, 14 September 2022

Dear Reviewer,

We are very happy to resubmit the revised version of the manuscript entitled “Spatial Prioritization for Ecotourism Through Applying the Landscape Resilience Model”.

We appreciate you for your valuable comments and suggestions which improved the manuscript considerably. We have taken all of your comments seriously and added new text to the manuscript as requested. Minor changes such as the correction of typos are not listed. We also sent the paper to a professional language review in England.

The revised manuscript has been prepared according to all the comments and views of the reviewer in green color highlight and is attached as a “Revised Manuscript”. We have also included a point-by-point response to the comments in addition to making the changes described above in the manuscript.

We hope this current version would be suitable for publication in the Journal of Land, and please do not hesitate to contact us if any further information would be required.

Reviewer 2:

  1. It is a very interesting topic to identify priority areas for sustainable ecotourism development through ecotourism potential model and landscape resilience measurement.
  2. In the "Abstract" section, it is suggested to moderately expand the research conclusions.

# Done as suggested: Lines 25-27

  1. In the "Introduction" section, it is suggested to supplement the utility of ecotourism to tourists and their willingness to pay, as well as related research literature on tourists' resources and facilities in ecotourism areas, so as to provide more basis for ecotourism potential model indicators.

# Done as suggested: Line 64-70.

  1. In the part of "2-1- Study Area", it is suggested to supplement climatic data such as air pressure, temperature, humidity, wind direction and speed, precipitation, thunderstorms, fog, radiation, cloud cover and cloud shape, as well as soil, geology, plants, animals and main crops, so as to provide sufficient background information for eco-tourism research. In addition, "Fig. 1." correctly places the scale position and improves the resolution of the drawing.

# Done as suggested: Line 85-94 & Line 98-99

  1. If possible, provide real photos of typical Eco-tourism areas in the study area so that readers can better understand them.

# Done as suggested: Fig. 1

  1. In the section of "3- Results", it is suggested to adjust the map layout moderately, so that Figures 5, 6, 7 and 8 are of the same size, which is easy to compare and read.

# Done as suggested

  1. It is suggested to add a "discussion"section. Based on the existing research results and existing research literature, analyze, discuss and extrapolate 2-3 topics to put forward the basis for the research conclusion.

# Done as suggested

  1. Research conclusions should be generalized and of general significance on the basis of case studies. Suggestions to deepen.

# Done as suggested: Lines 360-362

  1. It is also suggested to supplement the research deficiencies and further research directions.

# Done as suggested: Line 363

Reviewer 3 Report

Dear Author, 

Thank you for your research article, and I think this is a remarkable idea and analysis skills. But as an SSCI-indexed journal, we need more Author's INTERPRETATION as a social aspect. The current form is much like an analysis report with specific software. Please reduce the broad and variable colorful figure, but concentrate on the meaning and interpretation. Thank you.

1. We need more detailed and specific results data in the abstract. The results part of the current abstract is too vague.

2. Even though the high-resolution map in Fig. 1 (left map) is hard to match with the different color legends and map, please enlarge the map of fig1.

3. Fig2. It is hard to recognize the tiny font size in the table and equation model in the flowchart. Please revise the whole figure more clearly with high resolution, and delete the mark on the left side and end of each sentence.

4. We need more detailed information and characteristics of the ten experts, ex., their job type, job experience year, work field, gender, etc., and we need the survey or interview time too (When did the survey proceed and how long?) 

5. The experts' interview was a survey. Or an interview? Or was it mixed?

We need the experts' survey questionnaire or interview subjects, questions too. 

If it is too long, please add it as an Appendix.

6. Overall, please describe more detailed information in the expert survey.

(time, place, characteristics of experts, survey type, duration, etc.)

7. The format of figures and tables is different. Please make the same all figures and tables (format, design, size, etc.). Because a figure has been outlined, but some have not. The table format is different from that previous paper in LAND.

8. It is hard to find the logic flow from fig.4 to fig.5. How can the author produce the figure 5 result from figure 4? Please add more explanation.

9. The sizes of Fig 7 and Fig 5 are different.

10. Please revise figure 7. We can not recognize the seven protected areas with different color lines. Furthermore, the seven protected area lines are confused with the three categories of Zoning of the Biosphere reserve (core, buffer, transition zone)

11. What is different between figure 5 and figure 8? It looks pretty similar, and I wonder how the author resulted from both figures as other figures. 

Author Response

                                                                                     Isfahan, 14 September 2022

Dear Reviewer,

We are very happy to resubmit the revised version of the manuscript entitled “Spatial Prioritization for Ecotourism Through Applying the Landscape Resilience Model”.

We appreciate you for your valuable comments and suggestions which improved the manuscript considerably. We have taken all of the your comments seriously and added new text to the manuscript as requested. Minor changes such as the correction of typos are not listed. We added the supplementary data at the end of the manuscript. We also sent the paper to a professional language review in England.

The revised manuscript has been prepared according to all the comments and views of the reviewer in green color highlight and is attached as a “Revised Manuscript”. We have also included a point-by-point response to the comments in addition to making the changes described above in the manuscript.

We hope this current version would be suitable for publication in the Journal of Land, and please do not hesitate to contact us if any further information would be required.

Reviewer 3:

Thank you for your research article, and I think this is a remarkable idea and analysis skills. But as an SSCI-indexed journal, we need more Author's INTERPRETATION as a social aspect. The current form is much like an analysis report with specific software. Please reduce the broad and variable colorful figure, but concentrate on the meaning and interpretation. Thank you.

  1. We need more detailed and specific results data in the abstract. The results part of the current abstract is too vague.

# Done as Suggested: Line 20-23

  1. Even though the high-resolution map in Fig. 1 (left map) is hard to match with the different color legends and map, please enlarge the map of fig1.

# Done as suggested

  1. Fig2. It is hard to recognize the tiny font size in the table and equation model in the flowchart. Please revise the whole figure more clearly with high resolution, and delete the mark on the left side and end of each sentence.

# Done as suggested

  1. We need more detailed information and characteristics of the ten experts, ex., their job type, job experience year, work field, gender, etc., and we need the survey or interview time too (When did the survey proceed and how long?)

# Done as suggested: Line 146-149.

  1. The experts' interview was a survey. Or an interview? Or was it mixed?

We need the experts' survey questionnaire or interview subjects, questions too. 

If it is too long, please add it as an Appendix.

# Done as suggested

  1. Overall, please describe more detailed information in the expert survey.

(time, place, characteristics of experts, survey type, duration, etc.)

# Done as suggested. Regarding to the COVID-19 pandemic during our survey, we sent the prepared questionnaire to the selected experts and asked them to complete them. So, it is hard to say how long it takes for completing each questionnaire.

  1. The format of figures and tables is different. Please make the same all figures and tables (format, design, size, etc.). Because a figure has been outlined, but some have not. The table format is different from that previous paper in LAND.

# Done as suggested

  1. It is hard to find the logic flow from fig.4 to fig.5. How can the author produce the figure 5 result from figure 4? Please add more explanation.

# Done as suggested: Line: 155-158

  1. The sizes of Fig 7 and Fig 5 are different.

# Done as suggested

  1. Please revise figure 7. We can not recognize the seven protected areas with different color lines. Furthermore, the seven protected area lines are confused with the three categories of Zoning of the Biosphere reserve (core, buffer, transition zone)

# Done as suggested

  1. What is different between figure 5 and figure 8? It looks pretty similar, and I wonder how the author resulted from both figures as other figures. 

# Thank you very much for your useful comments. We changed the resulting map of Fig.9 to show the suitable zones for sustainable ecotourism development and make the results more transparent and robust.

Round 2

Reviewer 3 Report

- Thank you for your revision

- Table 2 shows very limited information with only the 'weight' value, please consider the merged table 2 with table 3

Author Response

                                                                                     Isfahan, 20 September 2022

Dear Reviewer,

We are very happy to resubmit the revised version of the manuscript entitled “Spatial Prioritization for Ecotourism Through Applying the Landscape Resilience Model”.

We appreciate you for your valuable comments and suggestions which improved the manuscript considerably. We have taken your comments seriously and merged table 2 and 3 together. We also sent the paper to a professional language review in England.

The revised manuscript has been prepared according to the comments and views of the reviewer in blue color highlight as well as marked up using the “Track Changes” and is attached as a “Revised Manuscript”. We have also included a point-by-point response to the comments in addition to making the changes described above in the manuscript.

We hope this current version would be suitable for publication in the Journal of Land, and please do not hesitate to contact us if any further information would be required.

Reviewer 3, 2nd Round:

Thank you for your revision

- Table 2 shows very limited information with only the 'weight' value, please consider the merged table 2 with table 3

# Done as Suggested: Line 207 & Table 2
